# Synthesis, Characterization and Antibacterial Application of Copolymers Based on *N*,*N*-Dimethyl Acrylamide and Acrylic Acid

**DOI:** 10.3390/ma14206191

**Published:** 2021-10-18

**Authors:** Ulantay Nakan, Shayahati Bieerkehazhi, Balgyn Tolkyn, Grigoriy A. Mun, Mukhit Assanov, Merey E. Nursultanov, Raikhan K. Rakhmetullayeva, Kainaubek Toshtay, El-Sayed Negim, Alibek Ydyrys

**Affiliations:** 1Institute of Geology and Oil & Gas Business, Satbayev University, Almaty 050013, Kazakhstan; merey1980@mail.ru; 2Ross Tilley Burn Centre, Sunnybrook Health Sciences Centre, University of Toronto, Toronto, ON M5S 1A1, Canada; sayahat2017@gmail.com; 3A.B. Bekturov Institute of Chemical Sciences, Street Ualihanov106, Almaty 050010, Kazakhstan; balgn-888@mail.ru; 4Al-Farabi Kazakh National University, Almaty 050012, Kazakhstan; mun-grig@yandex.ru (G.A.M.); muhit777.82@mail.ru (M.A.); raichan-rach@mail.ru (R.K.R.); kainaubek.toshtay@gmail.com (K.T.); 5Laboratory of Advanced Materials and Technology, Kazakh-British Technical University, 59 Tole bi St., Almaty 050000, Kazakhstan; elashmawi5@yahoo.com; 6Biomedical Research Centre, Al-Farabi Kazakh National University, al-Farabi 71, Almaty 050040, Kazakhstan; ydyrys.alibek@gmail.com

**Keywords:** *N*,*N*-dimethyl acrylamide, acrylic acid, reactivity ratios, Fineman–Ross, inverted Fineman–Ross, Kelen–Tudos, antibacterial activity

## Abstract

Hydrogel copolymers based on *N*,*N*-dimethyl acrylamide (DMA) and acrylic acid (AAc) were synthesized using a solution polymerization technique with different monomer ratios and ammonium persulfate as an initiator. This paper investigates the thermal stability, physical and chemical properties of the hydrogel copolymer. Testing includes Fourier transform infrared spectroscopy (FTIR), thermogravimetric analysis (TGA), scanning electron microscopy (SEM) and elemental analysis (CHNS). The copolymer composition was determined by elemental analysis, and the reactivity ratios of monomers were calculated through linearization methods such as Fineman–Ross (FR), inverted Fineman–Ross (IFR), Kelen–Tudos (KT) and Mayo–Lewis (ML). Good agreement was observed between the results of all four methods. The ratio of r_1_ and r_2_ were 0.38 (r_1_) and 1.45 (r_2_) (FR), 0.38 (r_1_) and 1.46 (r_2_) (IFR), 0.38 (r_1_) and 1.43 (r_2_) (KT), and 0.38 (r_1_) and 1.45 (r_2_) (ML). Hydrogel copolymers exhibited good thermal stability, and SEM showed three-dimensional porous structures. Antibiotic-free and antibiotic-loaded hydrogels demonstrated antimicrobial properties against both Gram-positive and Gram-negative bacteria. As the ratio of DMA in hydrogel copolymer increased, the activity of copolymer against bacteria enhanced. The results indicated that these hydrogels have the potential to be used as antibacterial materials.

## 1. Introduction

Three-dimensional polymers with unique properties have shown great promise in the polymer industry, with potential applications including medical appliances in drug delivery, electronics, biotechnology, and the food industry. Stimuli-responsive hydrogels belong to a new and vital class of cross-linked polymers [1,2,3,4,5].

Poly (*N*-isopropyl acrylamide)-based stimuli-responsive copolymers demonstrate great potential in biomedical applications. Poly(*N*-isopropyl acrylamide) affords phase separation caused by reversible hydration transition in aqueous solutions with temperatures higher than 32 °C. Polymers with similar remarkable capabilities include copolymers based on *N*,*N*-diethyl acrylamide, *N*,*N*-dimethyl acrylamide and N-vinyl caprolactam monomers. *N*,*N*-dimethyl acrylamide based hydrogels have gained significant attention due to their potential for many different applications. Studies on DMA based copolymers have reported excellent characteristics. However, the mechanical strength of these synthesized hydrogels is weak [6,7,8,9].

The copolymer composition and reactivity ratios of the monomer are crucial in evaluating copolymer specific applications. The reactivity monomer ratios calculated by conventional linear methods are not always accurate, but several non-linear ways have been developed to assess their value [10,11].

Polymers and polymer-based materials are not only widely used in the biomedical field, and they are also receiving more attention in food science and other technologies as well [12,13]. One of the crucial positions in the medical field is the antibacterial actions of the materials used in medicine, since bacterial growth must be prevented on medical devices, such as prosthetic materials, surgical masks, etc. In addition, the antibacterial functions are more desirable for preserving food quality and water sanitations as well.

Our lab prepared water swelling and soluble thermosensitive copolymer poly(*N*-isopropyl acrylamide-co-2-hydroxyl ethyl acrylate) and various physical and chemical methods have studied their different characteristics. Linear copolymer composition and reactivity monomer ratios were calculated by Kelen–Tudos and Fineman–Ross methods [14,15,16,17].

In this study, poly(DMA-co-AAc) hydrogels were synthesized by free-radical copolymerization. These hydrogels were characterized by CHN, FTIR, TGA, and SEM. In addition, the monomer reactivity ratios of DMA and AAc were determined via using Fineman–Ross (FR), inverted Fineman–Ross (IFR), Mayo–Lewis (ML) and Kelen–Tudos (KT) methods. Several biological tests were performed to determine these hydrogels antibacterial features and drug delivery capabilities.

## 2. Materials and Methods

### 2.1. Materials

Acrylic acid (AAc) stabilized with hydroquinone with a purity of 99.5% extra pure (CAS 79-10-7) was supplied by Acros Organics (Geel, Belgium). *N*,*N*-dimethyl acrylamide (DMA, 99%), ammonium persulfate (98%, APS, CAS 7727-54-0), and *N*,*N*-methylene-bis-acrylamide (99%, CAS 110-26-9) were obtained from Sigma-Aldrich (Heidelberg, Germany). Furthermore, the gentamicin of clinical grades was obtained from Kazakhstan in the specification of 1 mL:40 units (Shymkent, Kazakhstan, 40 mg/mL). In addition, several types of bacterial culture were obtained from the laboratory.

### 2.2. Synthesis of Copolymers

Hydrogels based on DMA and AAc were synthesized with composition ratios (M1: 30/70, M2: 40/60, M3: 50/50, M4: 60/40, M5: 70/30), by free radical solution (water) polymerization technique with ammonium persulfate (2 × 10^−2^ M) as the initiator and *N*,*N*-methylene-bis-acrylamide (0.1 mol.%) as the cross-linking agent. Monomer mixture accounts for 30% of the total volume, and the rest belongs to the water. The total volume of the reaction mixture was maintained at 10 mL for all the compositions. The resulting mixture was poured into the glass ampoule and saturated with argon for 10 min to remove oxygen. Then the copolymerization was performed in hermetically sealed glass ampoules at 60 °C for 20 min. The obtained hydrogels were washed with distilled water for 10 days to clean the samples from unreacted monomers. Then, the hydrogel samples were dehydrated at room temperature and in a vacuum oven until the weight was kept constant [15].

### 2.3. Instrumentation and Methods

Fourier transforms infrared spectrophotometer (FTIR) IR Nicolet 5700 spectrometer was used to analyze chemical structures of poly(DMA-co-AAc) copolymers in the 400–4000 cm^−1^ wavenumber range. The dried and grounded (until suitable sized powder) hydrogel samples mixed with standard KBr powder were compressed into a pellet for the test.

To describe the composition of poly(DMA-co-AAc) copolymer by elemental analysis method, the dried hydrogel samples were carried out using 2400 CHNS/O Series II System by Perkin Elmer elemental carbon, hydrogen, nitrogen and sulfur components in the samples were measured by weight percentage.

Hydrogels thermal properties were determined by thermogravimetric analysis (TGA) between 30 and 600 °C with a 10 °C/min heating rate under nitrogen atmosphere on a Perkin Elmer Simultaneous Analyzer STA 6000 (PerkinElmer, Waltham, MA, USA).

The surface morphology of hydrogel was investigated by scanning electron microscopy (SEM), on JSM-6390LV, (JEOL, Tokyo, Japan), with an operating voltage of 20 kV. SEM images were obtained from the fractured surface of dried hydrogels.

### 2.4. Monomer Reactivity Ratio

The monomer reactivity ratios were determined by using the Fineman–Ross (FR), inverted Fineman–Ross (IFR), Kelen–Tudos (KT) and Mayo–Lewis methods [18,19,20,21]. The equation for IFR is:G = Hr_1_ − r_2_
r_1_ and r_2_ were calculated by plotting G = F(f − 1)/f versus the H = F^2^/f to obtain a straight line where the slope was the value for (r_1_) and the intercept was the value for (−r_2_). The values are shown in plots of IFR and FR.

For KT method, the equation is:η = r_1_ ζ − r_2_/a(1 − ζ)

Where, η = G/(α + H), ζ = H/(α + H), α = (H_max_ * H_min_)^1/2^
r_1_ and r_2_ were calculated by plotting η versus ζ, and a straight-line was produced which gave (−r_2_/α) and (r_1_). The values are shown in KT plot. For the Mayo–Lewis method, a plot was obtained by using the equation:r_2_ = F [1/f(1 + Fr_1_) - 1](1)

### 2.5. Antibacterial Tests

The antibacterial and drug delivery properties of the poly(DMA-co-AAc) hydrogels with different feed compositions were determined on nutrient agar. For this purpose, the obtained hydrogels were washed with distilled water for several days to remove unreacted monomers. Then, the hydrogel disks (diameter of 7–8 mm and height of 1 mm) were dehydrated in a vacuum oven at room temperature until the weight remained constant. Two hydrogel disks from each of the M1, M3 and M5 hydrogels samples were cleaned and dried. One was immersed in a solution and the other was in an antibiotic solution (Gentamicin). All samples were stored at room temperature for 24 h. The hydrogel disks retained their soft form, absorbing the necessary solutions.

We chose *Staphylococcus aureus* as the test organism, which is a Gram-positive (GP) bacterium. GP bacteria can be colored crystal violet due to a thick layer of peptidoglycan. In contrast, Gram-negative (GN) bacteria can be turned red or pink. A variety of inflammatory diseases, including skin inflammation, pneumonia, sepsis and other infections, can be caused by *Staphylococcus aureus* [15].

Antimicrobial tests of poly(DMA-co-AAc) hydrogels were studied using GP bacteria (*Staphylococcus aureus* ATCC 6538-p) and GN bacteria (*E. coli* (ATCC 25922) and *P. aeruginosa* (ATCC 9027)) using the disc diffusion method. The composition (for *staphylococcus aureus*) of the nutritional agar was as follows (g/L): meat extract 1.5; sodium chloride 5, yeast extract 1.5; peptone 5; agar 15.0; pH 7.4–7.6. Incubation of test organisms with applied copolymers was carried out in the thermostat (Binder) at 37 °C for 24 and 48 h. The antimicrobial activity of the hydrogels was determined by measuring the diameter (in mm) of their inhibition zones to microorganisms’ growth. Photo documents confirmed the results of the studies.

## 3. Results and Discussion

### 3.1. FTIR Analysis of Poly(DMA-co-AAc) Hydrogel

The infrared spectroscopy method was utilized to confirm the functional groups in the copolymers. Although FTIR is a conventional way of studying copolymers, it has not lost its relevance in modern research. Infrared spectroscopy was applied to confirm the structure of the poly(DMA-co-AAc) copolymer. The obtained copolymers were recorded on FTIR spectroscopy in 500–4000 cm^−1^ as shown in Figure 1. Peaks at 2924, 2848, 2908 and 2843 cm^−1^ for C-H groups (stretching, aliphatic), 1670, 1608, 1586, and 1548 cm^−1^ for group C-N 3500–3200 cm^−1^, 1790 cm^−1^ for group C=O (group ester), 3500–3200 cm^−1^ for OH group were recorded [22,23,24]. These signals expressed the structural formula of the poly(DMA-co-AAc).

### 3.2. Monomer Reactivity Ratio

The copolymer composition is based on monomers and element feed. Therefore, it is essential to know monomers and radicals’ reactivity in describing the polymerization process. Although the poly(DMA-co-AAc) hydrogels were obtained during the synthesis by the initial monomeric composition, the calculations were performed using element analysis results. One of the poly(DMA-co-AAc) copolymer features is the presence of nitrogen in a single monomer, which allows calculating the copolymer content through the proportion of nitrogen. Thus, we made calculations based on nitrogen. Table 1 shows the influence of the molar fraction of DMA in copolymer (m_1_), the mole fraction of DMA monomer in (M_1_) and nitrogen from elemental analysis. The reactivity ratio was investigated by using Fineman–Ross (FR), inverted Fineman–Ross (IFR), Kelen–Tudos (KT) and Mayo–Lewis methods. Table 1 shows the parameters of the FR, IFR and KT equations. The reactivity ratio for DMA (r_1_) and AAc (r_2_) from the FR and IFR plot (Figure 2), FT and ML (Figure 3) plot are given in Table 2. The molar fraction of the DMA in the copolymer versus its molar fraction in the feed is shown in Figure 4.

KT, IFR, FR and ML calculations showed that the values of r_1_ and r_2_ obtained by these methods were associated with each other and were based on the composition diagram copolymer and the presence of r_1_ < 1 and r_2_ > 1. Under the influence of polar and spatial factors in the monomer and radical, i.e., the product of constants is less than one (r_1_* r_2_ < 1), which follows the monomer links of AAc and DMA tend to alternate in the chain

### 3.3. Thermogravimetric Analyses (TGA)

Thermal stability and the DMA ratio affecting the weight loss in hydrogels were investigated by TGA/DTG at 29–594 °C with N_2_ in the inert atmosphere. TGA thermogram for poly(DMA-co-AAc) hydrogels (M3 and M5) are shown in Figure 5 and Table 3. The thermal degradation for copolymers (M3 and M5) proceeds in three stages. The first weight loss (12.576 wt %) occurred between 29 and 160 °C for M3, (6.76 wt %) was between 30.24 and 209.6 °C for M5 to removing water and moisture. Furthermore, the weight loss increased from 6% to 12%, increasing the ratio of AAc from 30% to 50% in the copolymer, indicating that polyacrylic acid (PAAc) hydrogels could absorb a large amount of water without dissolving [25]. The second and third stages occurred between 160.31 °C and 594 °C for M3 and between 209.6 and 594.5 °C for M5, which was related to the decomposition with losing carboxyl, amid and hydroxyl groups in the copolymers. Thermal stability and initial degradation temperature were at 380 °C, as shown by DTG, demonstrating the stable characteristic of the poly(DMA-co-AAc) copolymers.

### 3.4. Scanning Electron Microscopy (SEM)

One of the crucial properties of the hydrogels is their surface morphologies, which allow characterizing the properties and prospects of the copolymer for further research and applications. Figure 6 shows the SEM of poly(DMA-co-AAc) hydrogels (M3). The surface morphologies verify that the prepared hydrogels have three-dimensional porous structures. A three-dimensional structure might be a better cross-link of the hydrogel, and pores and permeable surfaces in hydrogels can increase swelling capacity. The interaction of water molecules with either hydrophilic groups or water permeation increases the hydrogel’s porous structure that is the principal reason for the higher swelling ratios.

### 3.5. Antibacterial Abilities of the Hydrogels Immobilized with the Drugs

The antibacterial and drug delivery property of poly(DMA-co-AAc) hydrogels were evaluated, which might have the potential for biological applications and the use on surfaces for many applications and environments. GP bacteria [*Staphylococcus aureus* (ATCC 6538-p)] and two GN bacteria (*E. coli* (ATCC 25922) and *P. aeruginosa* (ATCC-9027)) were used for this purpose. Dry poly(DMA-co-AAc) hydrogels were immersed into gentamicin or water to absorb the liquid and the drug-free hydrogels were used as a control.

It was established that antibiotics diffused well into the medium from the carriers, inhibiting the test microorganism’s growth and creating a lysis zone around the test carrier with antibiotics, indicating the drug delivery traits of the hydrogels, shown in Figure 7 and Figure 8. Interestingly, a clear inhibition zone was also observed by hydrogels without antibiotic solutions, suggesting the antibacterial role of the hydrogels against GN bacteria *E. coli* and *P. aeruginosa*. However, the antibiotic-free poly(DMA-co-AAc) hydrogel did not show any remarkable bacteriostatic impact on the GP bacteria (ATCC 6538). In both cases, hydrogels with gentamicin showed promising results in suppressing bacterial growth, exhibiting a clear inhibition zone in GN and GP. This activity has the potential for important future applications, bearing in mind that high drug delivery ability with excellent antibacterial efficiency is a good feature for biological and medical applications.

## 4. Conclusions

New poly(DMA-co-AAc) hydrogel was synthesized with different composition ratios of *N*,*N*-dimethyl acrylamide and acrylic acid by free-radical copolymerization using ammonium persulfate as an initiator. Different techniques have studied the physiochemical properties of hydrogel copolymer. TGA technique used to study the thermal stability of hydrogel copolymers. CHN was the main technique and linear methods to determine the copolymer composition and reactivity ratios Fineman–Ross (r_1_ = 0.38; r_2_ = 1.45), inverted Fineman–Ross (r_1_ = 0.38; r_2_ = 1.46), Kelen–Tudos (r_1_ = 0.38; r_2_ = 1.43), and Mayo–Lewis (r_1_ = 0.38; r_2_ = 1.45). As shown in the results, the calculated monomer reactivity ratios showed higher reactivity for AAc compared to DMA. Therefore, the growing radicals showed a higher tendency to react with AAc monomer due to the higher reactivity of AAc. SEM showed characteristics of poly(DMA-co-AAc) polymer network. poly(DMA-co-AAc) hydrogel without the addition of gentamicin did not show antibacterial activity against *Staphylococcus aureus*. poly(DMA-co-AAc) hydrogels with gentamicin showed an ability to inhibit the growth of *Staphylococcus aureus*. However, an increase in the molar ratio of DMA in the copolymer causes an increase in the ability of copolymer to reduce the growth of *Staphylococcus aureus*. Overall, future perspectives in developing safe and effective new hydrogels with drug delivery properties especially involving co-delivery of antimicrobial polymers and conventional antimicrobial agents for medical and hygienic applications.

## Figures and Tables

**Figure 1 materials-14-06191-f001:**
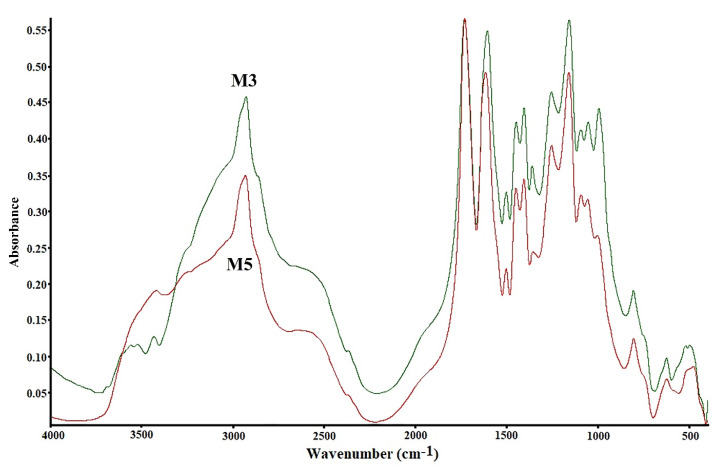
FTIR signals of hydrogels poly(DMA-co-AAc).

**Figure 2 materials-14-06191-f002:**
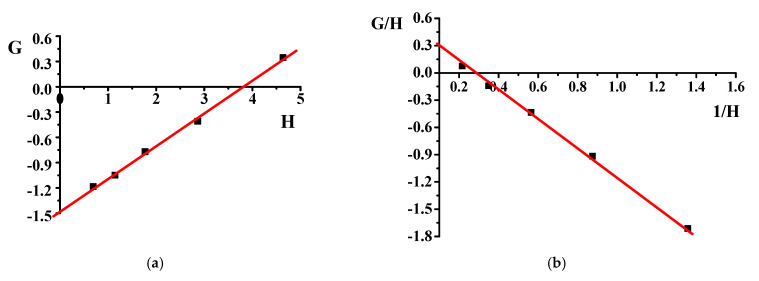
Fineman–Ross (**a**) and inverted Fineman–Ross (**b**) plots of poly(DMA-co-AAc).

**Figure 3 materials-14-06191-f003:**
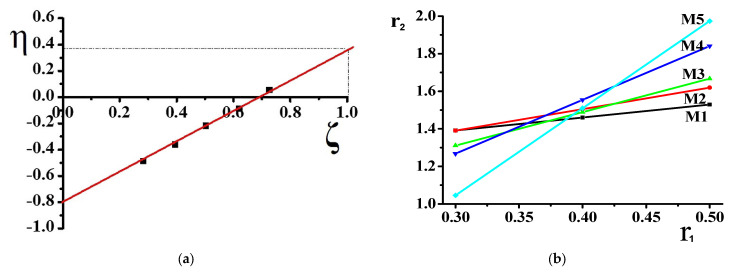
Kelen–Tudos (**a**) and Mayo–Lewis (**b**) plot of poly(DMA-co-AAc) copolymer.

**Figure 4 materials-14-06191-f004:**
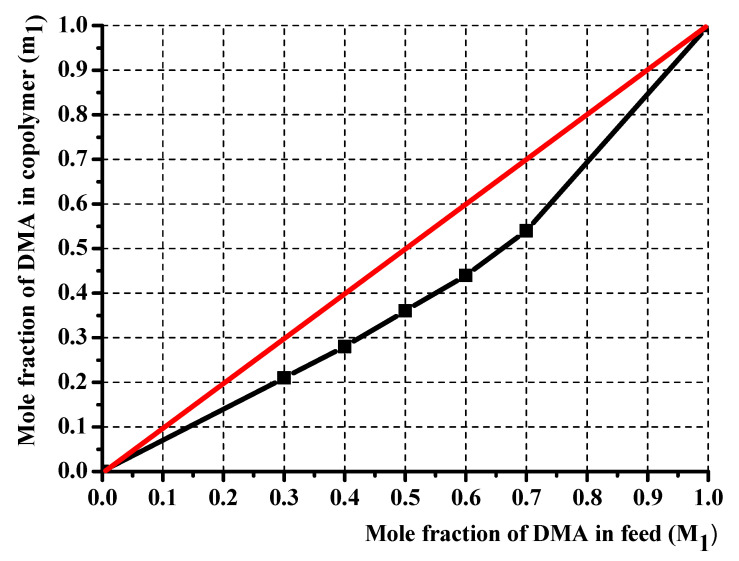
Composition of poly(DMA-co-AAc) copolymer.

**Figure 5 materials-14-06191-f005:**
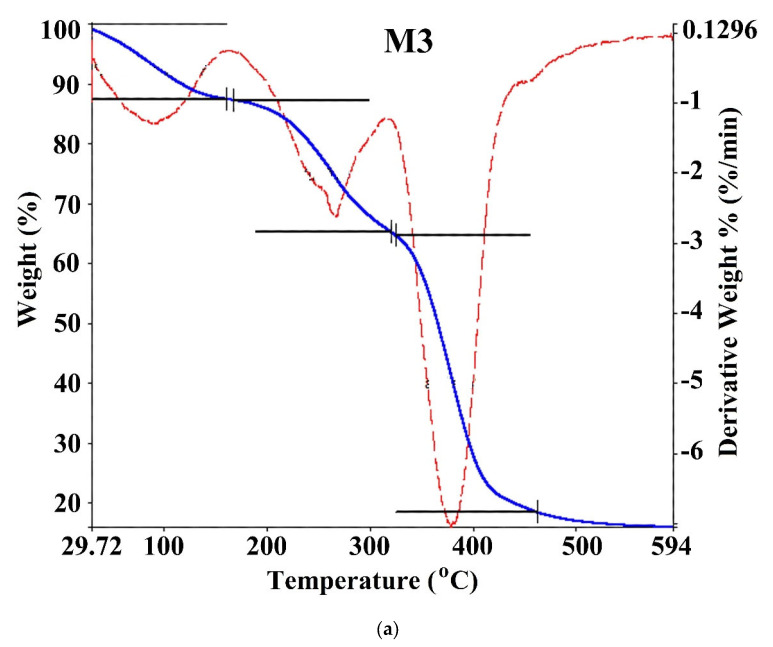
TGA curves of poly(DMA-co-AAc) hydrogel ((**a**) for M3 and (**b**) for M5).

**Figure 6 materials-14-06191-f006:**
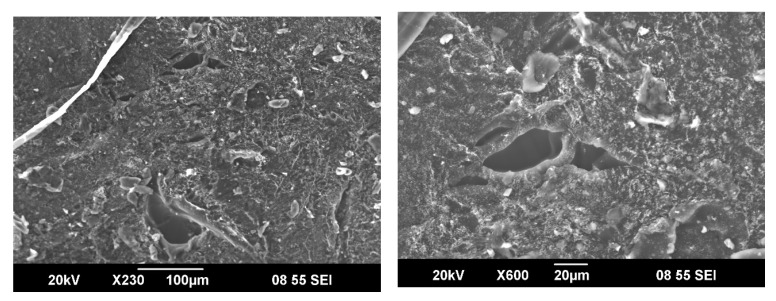
SEM Micrographs of poly(DMA–co–AAc) hydrogel (M3).

**Figure 7 materials-14-06191-f007:**
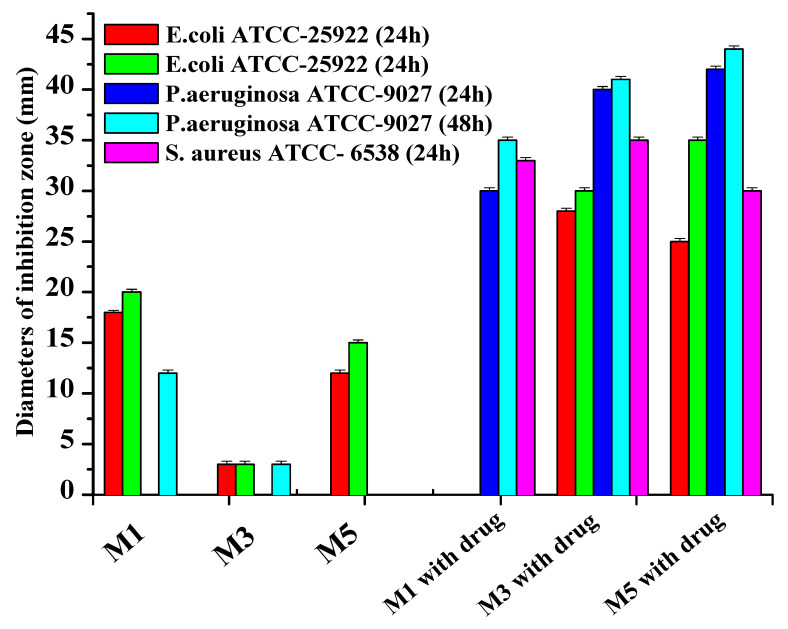
Inhibition zones of poly(DMA-co-AAc) hydrogels against GP (*Staphylococcus aureus*) and GN (*E. coli and P. aeruginosa bacteria*).

**Figure 8 materials-14-06191-f008:**
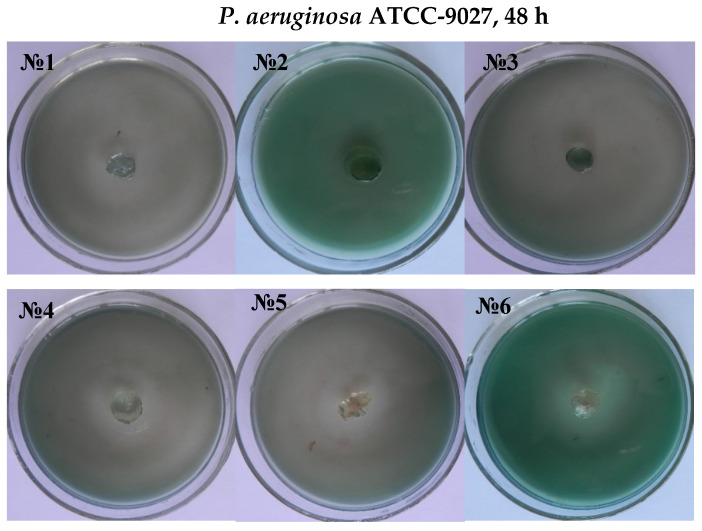
Antimicrobial activity of antibiotic-free and antibiotic-loaded poly(DMA-co-AAc) hydrogels against GP and GN bacteria (№ 1, 2, 3 for M1, M3, M5; № 4, 5, 6 for M 1, M3, M5 with gentamicin).

**Table 1 materials-14-06191-t001:** Determination of copolymerization reactivity ratios of DMA (r_1_) and AAc (r_2_) systems by Fineman–Ross, inverted Fineman–Ross and Kelen–Tudos methods.

№ Copolymer	Mole Fraction of DMA in Feed, (M_1_)	Elemental Analysis N, %	Mole Fraction of DMA in Copolymers, (m_1_) *			Fineman–Ross		Kelen–Tudos		Inverted Fineman–Ross	
				F = M_1/_M_2_	f = m_1/_m_2_	G = F(f−1)/f	H = F^2^/f	η = G/(α + H)	ζ = H/(α + H)	G/H	1/H
M1	30	3.85	21	0.4286	0.2658	−1.1839	0.6911	−0.4885	0.2831	−1.7131	1.4469
M2	40	4.93	28	0.6667	0.3889	−1.0476	1.1429	−0.3621	0.3945	−0.9166	0.8749
M3	50	6.167	36	1	0.5625	−0.778	1.778	−0.2189	0.5029	−0.4353	0.5647
M4	60	7.37	44	1.5	0.7857	−0.4091	2.8636	−0.089	0.6210	−0.1429	0.3492
M5	70	8.71	54	2.3333	1.1739	0.3457	4.6378	0.0541	0.7261	0.0745	0.2156

* Calculated from nitrogen (N, %) results in copolymers. A = (H_max_ * H_min_)^1/2^ = 1.7499.

**Table 2 materials-14-06191-t002:** Copolymer constants based on Fineman–Ross, inverted Fineman–Ross, Kelen–Tudos and Mayo–Lewis methods.

№	Methods	r_1_	r_2_	r_1_*r_2_
1	Fineman–Ross (FR)	0.38	1.45	0.55
2	Inverted Fineman–Ross (IFR)	0.38	1.46	0.55
3	Kelen–Tudos (KT)	0.38	1.43	0.54
4	Mayo–Lewis	0.38	1.45	0.57

**Table 3 materials-14-06191-t003:** Thermal properties of poly(DMA-co-AAc) hydrogels.

Copolymer	Temperature Decomposition (°C)	Weight Loss (%)	Residual (%)	PDT_max_ (°C)
M3	29.72–160.31	12.576	87.624	380.92
	160.31–322	22.424	65	
	322–594	49	16	
M5	30.24–209.6	6.76	93.24	378.12
	209.6–317.36	21.61	71.63	
	317.36–594.5	58.71	12.92	

## Data Availability

The data presented in this study are available on request from the corresponding author.

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
