# Peer review of "Synthesis, Characterization and Antibacterial Application of Copolymers Based on N,N-Dimethyl Acrylamide and Acrylic Acid"

_materials, 2021, doi:10.3390/ma14206191_

Round 1

Reviewer 1 Report

Dear Authors ,

Hope you are doing well.

I have gone through your article, it was quite nice  but some minor corrections are needed which will enhance the quality of your article.

Please go through the recommended points below.

  1. Abstract need to be rewrite , there is no end result in the abstract.
  2. Introduction need to be improved. It should co-relate with the objective.
  3. Rational of the study need to be clear enough for the readers.
  4. It willl be great if authors could explain the amount of chemicals have been fixed for experiment such as o.1 M % of cross linking agent.
  5. Change disco-diffusion to Disc- diffusion (Line no 144).
  6. Change 370C to 37C.
  7. FTIR figure is not clear, insert different peak in spectra.
  8. Poor figure no.5, try to make a single figure with good resolution.
  9. Please maintain uniformity ,in some places author has mentioned as figure where as in another place has written fig.
  10. It would have been great if authors could show the zone of inhibition in appropriate manner
  11. E.Coli picture no-4 is not available.
  12.  Zone of inhibition of S.aureus is not clear.
  13. Conclusion is not clear and is not matched with the result, please rewrite the conclusion.

Author Response

September 12. 2021

Dear Prof.

We thank the Editors and the Reviewers for their thoughtful review of this manuscript and for their specific suggestions that are extremely instrumental to our efforts to improve the quality of this manuscript. We have revised the manuscript based on your kind advice and the referee’s detailed suggestions. Enclosed, please find the responses to the referees. We sincerely hope this manuscript will be finally acceptable to be published in Polymers. We do not want to publish this article in a special issue. We want it will be published in the regular issue.

Thank you very much and looking forward to hearing from you soon.

Best regards

Sincerely yours

Dr. Ulantay Nakan

Please find the following response to the comments of referees:

 Response to the referee’s comments

Reviewer #1

 Comment 1:     Abstract need to be rewrite, there is no end result in the abstract.

Response: The authors would like to thank deeply the reviewer for his time and effort.

The revised details can be found in the Abstract. We have corrected and marked it red in the manuscript.

Comment 2: Introduction need to be improved. It should co-relate with the objective Response: Thank you very much for your suggestion.

We have made substantial changes in introduction part of the paper to address the reviewer’s comment.

Comment 3: Rational of the study need to be clear enough for the readers.

Response: Thank you very much for your suggestion. We have made major revision and tried our best to improve our manuscript. We hope it can be meet with your approval.

Comment 4: It will be great if authors could explain the amount of chemicals have been fixed for experiment such as o.1 M % of cross- linking agent.

Response: Thanks for your advice. 

The manuscript states that the total volume of all compositions at 10 ml. 30% of this volume consists of monomers mixture and the rest of all volume is water used as a solvent (in the amount of 70%).  The monomer mixture is calculated based on the proportion of monomers ratio in feed.  for example, when the proportion of two monomers is the same in the feed (M3) their volumes used for synthesis are equal to each other. Amount of the crosslinking agent and the initiator have calculated by Mole fraction of monomers in feed.

Comment 5: Change disco-diffusion to Disc- diffusion (Line no 144).

Response: Thank you very much for your comment and suggestion.

We do agree with the points you raised. We have corrected error and marked it red in the manuscript.

Comment 6: Change 370C to 37°C.

Response: Thank you for pointing out that mistake. We checked and noticed that it is a typing mistake. We have corrected an error and marked it red in the manuscript.

Comment 7: FTIR figure is not clear, insert different peak in spectra.

Response: We appreciate for this suggestion.

FTIR figure has been corrected to conform to all standards.

Comment 8: Poor figure no.5, try to make a single figure with good resolution.

Response: Thanks for your suggestion.

We improved figures and the discussion of TGA.

The revised details can be found in section TGA (red color).

Comment 9: Please maintain uniformity, in some places author has mentioned as figure where as in another place has written fig.

Response: Thank you very much for your comment and suggestion!!! 

Correction has been made accordingly and it marked it red in the manuscript. (Fig 7 and 8. Figures 7 and 8.)

Comment 10: It would have been great if authors could show the zone of inhibition in appropriate manner

Response: We appreciate for your kind advice and suggestion.

We revised our manuscript and implicated our results with a proper manner, the revised version can be found in the second paragraph of results section 3.5 (shown in red color).

Comment 11: E.Coli picture no-4 is not available.

Response: We appreciate for your comments and suggestions!!!

Cleaned and dried hydrogel samples were immersed in the water solution and solution with antibiotics (Gentamicin). All samples are stored at RT for 24 hours, which allows the hydrogel disks maintain their soft form, absorbing the necessary solutions. However, sometimes hydrogels spontaneously crack while absorbing the solution, which happened in our case, sample No 4, making us unable to provide the photo. However, we believe that unavailability of Ecoli picture will not affect the conclusions.  

Comment 12: Zone of inhibition of S.aureus is not clear.

Response: Thanks for your suggestion.

We have already revised this part regarding your concern.

We have revised this part in our manuscript which is shown I red color.  Hydrogels without antibiotics against GP bacteria S.aureus showed no any inhibitory activity.

Comment 13:  Conclusion is not clear and is not matched with the result, please rewrite the conclusion.

Response: Thank you very much for your comment and suggestion.

We revised the conclusion.

Reviewer 2 Report

I read the manuscript, but there are many problems related to grammar, quality of the figures… The main problem is the discussion; authors do not use literature in many cases and just exhibit the results. I think the paper must identify a clear goal.

Some specific problems:

Grammar and format must be checked.

The title should be changed incorporating both monomers.

“Our lab prepared and evaluated hydrophilic and swelling thermosensitive copoly-50mer NIPAAm-2HEA via different physical and chemical methods. Linear copolymer 51composition and reactivity monomer ratios were calculated by Kelen-Tudos and Fine-52man-Ross methods [12-15].53” Please, do not use abbreviations, write the complete name.

Please, provide the purity of the reagents

Synthesis of copolymers: Please, include a table with all the amounts used for the synthesis including also the amount of solvent. That information is necessary for the evaluation of the work. The synthetic route must be better explained.

FTIR analysis of DMA-co-AAc hydrogel must include literature for supporting the data. The figure is too bad, please I think the results should be cleaned. You need to use NMR for composition of the random copolymers…

3.2.Monomer reactivity ratio: Authors must explain the calculations, and provide references. What is the error of those values?

Figure 2 and 3 are too bad.

Thermogravimetric Analyses (TGA): The discussion must be improved.

I read all the manuscript but problems are related to the explanation of the results… Discussion is poor.

Author Response

September 12. 2021

Dear Prof.

We thank the Editors and the Reviewers for their thoughtful review of this manuscript and for their specific suggestions that are extremely instrumental to our efforts to improve the quality of this manuscript. We have revised the manuscript based on your kind advice and the referee’s detailed suggestions. Enclosed, please find the responses to the referees. We sincerely hope this manuscript will be finally acceptable to be published in Polymers. We do not want to publish this article in a special issue. We want it will be published in the regular issue.

Thank you very much and looking forward to hearing from you soon.

Best regards

Sincerely yours

Dr. Ulantay Nakan

Please find the following response to the comments of referees:

 Response to the referee’s comments

  Reviewer #2

Comment 1.  Grammar and format must be checked.

Response: Thank you very much for your comments and suggestions.

we revised the grammar and language of the manuscript.

The revised details can be found in the text (red color)

Comment 2. The title should be changed incorporating both monomers.

Response: We appreciate this comment and thank you for this suggestion. 

We are grateful for your opinion on the title of this manuscript that you shared with us. We want this manuscript to have the following title. «Synthesis, Characterization and anti-bacterial application of copolymers based on N,N-dimethylacrylamide and acrylic acid».

We marked these changes in red.

Comment 3. “Our lab prepared and evaluated hydrophilic and swelling thermosensitive copoly-50mer NIPAAm-2HEA via different physical and chemical methods. Linear copolymer 51composition and reactivity monomer ratios were calculated by Kelen-Tudos and Fine-52man-Ross methods [12-15].53” Please, do not use abbreviations, write the complete name.

Response: Thanks for your suggestion.

We agree with the reviewer and have made the revision.

Comment 4. Please, provide the purity of the reagents.

Response: Thank you very much for your comments.

We do agree with the points you raised and we marked these changes in red.

Comment 5. Synthesis of copolymers: Please, include a table with all the amounts used for the synthesis including also the amount of solvent. That information is necessary for the evaluation of the work. The synthetic route must be better explained.

Response: We appreciate this comment and thank you for this suggestion.

The manuscript states that the total volume of all compositions at 10 ml. 30% of this volume consists of monomers mixture and the rest of all volume is water used as a solvent (in the amount of 70%).  The monomer mixture is calculated based on the proportion of monomers ratio in feed.  for example, when the proportion of two monomers is the same in the feed (M3) their volumes used for synthesis are equal to each other. Amount of the crosslinking agent and the initiator have calculated by Mole fraction of all monomers in feed.

Comment 6. FTIR analysis of DMA-co-AAc hydrogel must include literature for supporting the data. The figure is too bad, please I think the results should be cleaned. You need to use NMR for composition of the random copolymers…

Response: Thank you very much for your suggestion.

We included references for supporting the data (FTIR section).

NMR test is not available for my samples. All samples (hydrogels) are insoluble in solvents (DMSO-d6, D2O and other). They swell in the solution. NMR test is not available at our university. Also, we cannot send samples to another country because of Covid-19.

Comment 7. 3.2. Monomer reactivity ratio: Authors must explain the calculations, and provide references. What is the error of those values?

Response: Thank you very much for your comment and suggestion.

We explained the calculation of monomer reactivity.

The revised details can be found in section 4.2 (red color).

Also provided references, can be found in references (red color)

Comment 8: Figure 2 and 3 are too bad.

Response: Thank you very much for your advices.

Figure 2 and 3 have been improved to conform to all standards.

Comment 9: Thermogravimetric Analyses (TGA): The discussion must be improved.

Response: Thanks for your suggestions.

We improved the discussion of TGA.

The revised details can be found in section TGA (red color).

Comment 10: I read all the manuscript but problems are related to the explanation of the results Discussion is poor.

Response: Thank you very much for your comment and suggestion.

We improved the discussion. The revised details can be found in red color.

Reviewer 3 Report

The work deals with synthesizing of N, N-dimethyl acrylamide (DMA) and acrylic acid (АAc) copolymers with various DMA/AAc ratios. Prepared hydrogel copolymers were analyzed by FTIR, TGA, SEM and CHNS. Their antibacterial properties were studied by using both gram positive and negative bacteria strains.

However, there are some major concerns besides minor ones, as below:

Major comments:

C1) The authors mentioned two polymerization techniques of copolymerization and free radical polymerization. Samples prepared by free radical polymerization were named M1 to M5, according to the DMA ratio, and the results of some of them were presented in the manuscript. However, samples prepared by the copolymerization technique are not seen in the manuscript.

Authors made any analysis regarding the samples prepared by copolymerization?

C2) Most of the result does not include all samples, but some of the selected samples are shown, and others are missing. Please, include the results of all the samples of M1 to M5 for every section.

C3) Drug immobilization part is missing in the experimental part. Please mention about the immobilization of the drugs.

Minor comments:

C4) Line 23: Please indicate r1 and r2 here, as it was between lines 148-149.

C5) Line 33 : Please remove one of the “three” from the sentence.

C6) Line 44: Please add the references of many studies at the end of the sentence.

C7) Line 51: Authors named the sample as NIPAAm-2HEA, however,the rest of the manuscript mention it as DMA-co-AAc hydrogels. Please use the same abbreviation in the whole manuscript. Also, what 2HEA stands for is not clear.

C8) Line 71: Please remove one of the “based on” from the sentence.

C9) Line 100 : Dried hydrogel samples of No1-No6 were not mentioned here what is it stands for. It was given in Line 256, under Figure 8, however, it is supposed to be mentioned before using the abbreviation for the first time.

C10) Please be aware that each bacteria name is supposed to be given in italic font in the manuscript. None of them are italic.

C11) Line 116: Photo documents confirmed the results of the studies. Please indicate that its figure 8, which was showing the antimicrobial test results.

C12) Line 268: molar ration supposed to be molar ratio, please correct it.

C13) Line 268-269: Authors conclude that “However, an increasing in the molar ration of DMA in the copolymer, causes an increase in the ability of copolymer to reduce the growth of Staphylococcus aureus.”According to figure 7, there is no such linear effect of DMA against S. aureus.

Author Response

September 12. 2021

Dear Prof.

We thank the Editors and the Reviewers for their thoughtful review of this manuscript and for their specific suggestions that are extremely instrumental to our efforts to improve the quality of this manuscript. We have revised the manuscript based on your kind advice and the referee’s detailed suggestions. Enclosed, please find the responses to the referees. We sincerely hope this manuscript will be finally acceptable to be published in Polymers. We do not want to publish this article in a special issue. We want it will be published in the regular issue.

Thank you very much and looking forward to hearing from you soon.

Best regards

Sincerely yours

Dr. Ulantay Nakan

Please find the following response to the comments of referees:

 Response to the referee’s comments

Reviewer #3

  Major comments:

Comment C1) The authors mentioned two polymerization techniques of copolymerization and free radical polymerization. Samples prepared by free radical polymerization were named M1 to M5, according to the DMA ratio, and the results of some of them were presented in the manuscript. However, samples prepared by the copolymerization technique are not seen in the manuscript.

Authors made any analysis regarding the samples prepared by copolymerization?

Response: Thank you very much for your encouraging comments and suggestions.

DMA-co-AAc copolymer were synthesized from two different monomers by free radical polymerization. During synthesis, all the conditions of free radical polymerization were preserved. we changed " copolymerization and free radical polymerization" from "free radical copolymerization" in manuscript and marked these changes in red.

Comment C2) Most of the result does not include all samples, but some of the selected samples are shown, and others are missing. Please, include the results of all the samples of M1 to M5 for every section.

Response: We appreciate for your thoughtful suggestions.

We have provided all the samples (M1 to M5) for CHN analysis. The copolymer composition was determined by elemental analysis and calculated through linearization methods. If there are more points, it will be easier to drive straight and there will be fewer errors.

For other research methods, the most important ones were selected from all samples.

Comment C3) Drug immobilization part is missing in the experimental part. Please mention about the immobilization of the drugs.

Response: Thank you very much for your comment and suggestion.

The dried hydrogel samples were laced with Gentamicin solution and distilled water for 24 hours to absorb the antibiotics. There are uses the same value amount of water and antibiotic solution. These hydrogels belong to the class of thermosensitive polymers. DMA-co-AAc hydrogels allows the release of the drug solution by compression at 35-400C temperature range and allows transport within this temperature range. Then these discs are placed in a Petri dish that is bacteria infested. The bacteria isolate is grown mainly on the nutrient agar. After incubation overnight, the discs are observed whether that antibiotics hindered the growth of bacteria.

Minor comments:

Comment C4) Line 23: Please indicate r1 and r2 here, as it was between lines 148-149.

Thank you for your suggestion.

we indicated r1 and r2 individually in abstract. We marked these changes in red.

Comment C5) Line 33: Please remove one of the “three” from the sentence.

Response: Thank you very much for your comment and suggestion.

We have removed the errors in the introduction and marked these changes in red.

Comment C6) Line 44: Please add the references of many studies at the end of the sentence.

Response: We appreciate for your comments and suggestions.

We added some references. 

Comment C7) Line 51: Authors named the sample as NIPAAm-2HEA, however, the rest of the manuscript mention it as DMA-co-AAc hydrogels. Please use the same abbreviation in the whole manuscript. Also, what 2HEA stands for is not clear.

Response: Thank you very much for your suggestions.

We have marked these changes in red.

Comment C8) Line 71: Please remove one of the “based on” from the sentence.

Response: Thank you for your suggestion.

We have removed this word in our manuscript.

Comment C9) Line 100: Dried hydrogel samples of No1-No6 were not mentioned here what is it stands for. It was given in Line 256, under Figure 8, however, it is supposed to be mentioned before using the abbreviation for the first time.

Response: Thank you very much for your comment and suggestion.

We do agree with the points you raised and have marked these changes in red.

Comment C10) Please be aware that each bacteria name is supposed to be given in italic font in the manuscript. None of them are italic.

Response: Thanks for your advice.

We have marked these changes in red.

Comment C11) Line 116: Photo documents confirmed the results of the studies. Please indicate that its figure 8, which was showing the antimicrobial test results.

Response: We appreciate for your suggestion.

We revised our manuscript and implicated our results with an appropriate manner, the revised version can be found in the second paragraph of results section 3.5 (shown in red color).

Results of antimicrobial properties of antibiotic-free/loaded DMA-co-AAc hydrogels are shown in Figur 8 and Figur No7 (in the diagram).  Samples placed in petri dishes containing microorganisms are photographed after 24/48 hours.

Comment C12) Line 268: molar ration supposed to be molar ratio, please correct it.

Response: Thank for your suggestion.

We have marked these changes in red.

Comment C13) Line 268-269: Authors conclude that “However, an increasing in the molar ration of DMA in the copolymer, causes an increase in the ability of copolymer to reduce the growth of Staphylococcus aureus.” According to figure 7, there is no such linear effect of DMA against S. aureus.

Response: Thanks for your suggestion.

We have already revised this part regarding your concern.

Round 2

Reviewer 2 Report

I read the manuscript entitled:” Synthesis, Characterization and anti-bacterial application of co- 2 polymers based on N,N-dimethylacrylamide and acrylic acid” prepared by Ulantay Nakan, Shayahati Bieerkehazhi, Balgyn Tolkyn, Grigoriy A. Mun, Merey E. Nursultanov, Raikhan K., Rakhmetullayeva, Kainaubek Toshtay, El-Sayed Negim, Alibek Ydyrys.

The manuscript is slightly improved...

The quality of some figures needs to be addressed…

A clear explanation, and connection, between characterization techniques needs to be carried out. What is the purpose?

On the other hand, some suggestions and comments (previous revision) were not correctly addressed… Why? I need a clear explanation about those points.

Please, the discussion needs to be improved. What kind of information are you getting from characterization?

Author Response

October 02. 2021

Dear Prof.

We value the comments received greatly, as they have pointed out a number of important issues to be addressed. We would like to thank the editor and the reviewers for taking time in reading and suggesting modifications to the paper. We highly appreciate it, as the comments have been very useful to improve the paper. We improved to the initial manuscript based on the suggestions of the reviewers. We hope that the editor will find the paper suitable for publication.

Thank you very much for your kind consideration of this resubmitted version of our manuscript.

Best regards

Sincerely yours

Dr. Ulantay Nakan

Please find the following response to the comments of referees:

Response to the referee’s comments

Reviewer #2

 Comment.  The manuscript is slightly improved...

Response: Thank you very much for your suggestion.

We have made major revision and tried our best to improve our manuscript. We hope it can be meet with your approval.

Comment.  The quality of some figures needs to be addressed…

Response: We appreciate this comment and thank you for this suggestion. 

We have been improved the quality of Figures №1 to №5.

Comment.  A clear explanation, and connection, between characterization techniques needs to be carried out. What is the purpose?

Response: Thank you very much for your comments and suggestions.

We agree with the reviewer. The ‘Instrumentation and methods’ section has been rewritten with more clarity and specification by adding and deleting some sentences.  we showed the changes in the article and marked in the red

Comment. On the other hand, some suggestions and comments (previous revision) were not correctly addressed… Why? I need a clear explanation about those points.

Response: Thank you very much for your comments.

We have been rechecked and revised.

The revised details can be found in red color.

Comment . Please, the discussion needs to be improved. What kind of information are you getting from characterization?

Response: Thank you very much for your comments and suggestions.

We have been improved ‘discussion’ section. The revised details can be found in red color.

Reviewer 3 Report

The authors well addressed the comments regarding their manuscript. I am convinced that it can be published in its new form.

Author Response

October 02. 2021

Dear Prof.

We value the comments received greatly, as they have pointed out a number of important issues to be addressed. We would like to thank the editor and the reviewers for taking time in reading and suggesting modifications to the paper. We highly appreciate it, as the comments have been very useful to improve the paper. We improved to the initial manuscript based on the suggestions of the reviewers. We hope that the editor will find the paper suitable for publication.

Best regards

Sincerely yours

Dr. Ulantay Nakan